# COVID-19 and Arabic Twitter: How can Arab World Governments and Public Health Organizations Learn from Social Media?

**Lama Alsudias**
King Saud University / Saudi Arabia
Lancaster University / UK
`lalsudias@ksu.edu.sa`

**Paul Rayson**
Lancaster University / UK
`p.rayson@lancaster.ac.uk`

## Abstract

In March 2020, the World Health Organization announced the COVID-19 outbreak as a pandemic. Most previous social media related research has been on English tweets and COVID-19. In this study, we collect approximately 1 million Arabic tweets from the Twitter streaming API related to COVID-19. Focussing on outcomes that we believe will be useful for Public Health Organizations, we analyse them in three different ways: identifying the topics discussed during the period, detecting rumours, and predicting the source of the tweets. We use the k-means algorithm for the first goal with k=5. The topics discussed can be grouped as follows: COVID-19 statistics, prayers for God, COVID-19 locations, advise and education for prevention, and advertising. We sample 2000 tweets and label them manually for false information, correct information, and unrelated. Then, we apply three different machine learning algorithms, Logistic Regression, Support Vector Classification, and Naïve Bayes with two sets of features, word frequency approach and word embeddings. We find that Machine Learning classifiers are able to correctly identify the rumour related tweets with 84% accuracy. We also try to predict the source of the rumour related tweets depending on our previous model which is about classifying tweets into five categories: academic, media, government, health professional, and public. Around (60%) of the rumour related tweets are classified as written by health professionals and academics.

## 1 Introduction

The current coronavirus disease (COVID-19) outbreak is of major global concern and is classified by the World Health Organization as an international health emergency. Governments around the world have taken different decisions in order to stop the spread of the disease. Many academic researchers in various fields including Natural Language Processing (NLP) have carried out studies targeting this subject. For example, COVID-19 and AI[1] is one of the conferences that has been convened virtually to show how Artificial Intelligence (AI) can contribute in helping the Public Health Organizations during pandemics.

People use social media applications such as Twitter to find the news related to COVID-19 and/or express their opinions and feelings about it. As a result, a vast amount of information could be exploited by NLP researchers for a myriad of analyses despite the informal nature of social media writing style.

We hypothesise that Public Health Organizations (PHOs) may benefit from mining the topics discussed between people during the pandemic. This may help in understanding a population's current and changing concerns related to the disease and help to find the best solutions to protect people. In addition, during an outbreak people themselves will search online for reliable and trusted information related to the disease such as prevention and transmission pathways. COVID-19 Twitter conversations may not correlate with the actual disease epidemiology. Therefore, Public Health Organizations have a vested interest in ensuring that information spread in the population is accurate (Vorovchenko et al., 2017). For instance, the Ministry of Health in Saudi Arabia[2] presents a daily press conference incorporating the aim of quickly stopping the spread of false rumours. However, there is currently a prolonged period of time until warnings are issued. For example, the first tweet that included false information about hot weather killing the virus was on 10

---

[1] `https://hai.stanford.edu/events/covid-19-and-ai-virtual-conference`
[2] `https://www.moh.gov.sa`

February 2020, while the press conference which responded to this rumour on 14 April 2020. There is a clearly a need to find false information as quickly as possible. In addition, effort needs to be made in relation to tracking the user accounts that promote rumours. This can be undertaken using a variety of techniques, for example using social network features, geolocation, bot detection or content based approaches such as language style. Public Health Organizations would benefit from speeding up the process of tracking in order to stop rumours and remove the bot networks.

The vast majority of the previous research in this area has been on English Twitter content but this will not directly assist PHOs in Arabic speaking countries. The Arabic language is spoken by 467 million people in the world and has more than 26 dialects[3]. Of particular importance for NLP is coping with dialectal and/or meaning differences in less formal settings such as social media. As an example in health field, the word (تحصين) may be understood as vaccination[4] in Modern Standard Arabic or reading supplications in Najdi dialect[5]. There has been much recent progress in Arabic NLP research yet there is still an urgent need to develop fake news detection for Arabic tweets (Mouty and Gazdar, 2018).

In this paper, we have combined qualitative and quantitative studies to analyse Arabic tweets aiming to support Public Health Organizations who can learn from social media data along various lines:

- Analyzing the topics discussed between people during the peak of COVID-19

- Identifying and detecting the rumours related to COVID-19.

- Predicting the type of sources of tweets about COVID-19.

## 2   Related Work

There is a vast quantity of research over recent years that analyses social media data related to different pandemics such as H1N1 (2009), Ebola (2014), Zika Fever (2015), and Yellow Fever

---

[3]https://en.wikipedia.org/wiki/Arabic
[4]https://en.wikipedia.org/wiki/Modern_Standard_Arabic
[5]https://en.wikipedia.org/wiki/Najdi_Arabic

(2016). These studies followed a variety of directions for analysis with multiple different goals (Joshi et al., 2019). The study of Ahmed et al. (2019) used a thematic analysis of tweets related to the H1N1 pandemic. Eight key themes emerged from the analysis: emotion, health related information, general commentary and resources, media and health organisations, politics, country of origin, food, and humour and/or sarcasm.

A survey study (Fung et al., 2016b) reviewed the research relevant to the Ebola virus and social media. It compared research questions, study designs, data collection methods, and analytic methods. Ahmed et al. (2017b) used content analysis to identify the topics discussed on Twitter at the beginning of the 2014 Ebola epidemic in the United States. In (Vorovchenko et al., 2017), they determined the geolocation of the Ebola tweets and named the accounts that interacted more on Twitter related to the 2014 West African Ebola outbreak. The main goal of the study by Kalyanam et al. (2015) was to distinguish between credible and fake tweets. It highlighted the problems of manual labeling process with verification needs. The study in (Fung et al., 2016a) highlighted how the problem of misinformation changed during the disease outbreak and recommended a longitudinal study of information published on social media. Moreover, it pointed out the importance of understanding the source of this information and the process of spreading rumours in order to reduce their impact in the future.

Ghenai and Mejova (2017) tracked Zika Fever misinformation on social media by comparing them with rumours identified by the World Health Organization. Also, they pointed out the importance of credible information sources and encouraged collaboration between researchers and health organizations to rapidly process the misinformation related to health on social media. The study in (Ortiz-Martínez and Jiménez-Arcia, 2017) reviewed the quality of available yellow fever information on Twitter. It also showed the significance of the awareness of misleading information during pandemic spread. The study of Zubiaga et al. (2018a) summarised other studies related to social media rumours. It illustrated techniques for developing rumour detection, rumour tracking, rumour stance classification, and rumour veracity classification. Vorovchenko

et al. (2017) mentioned the importance of Twitter information during the epidemic and how Public Health Organisations can benefit from this. They showed the requirement to monitor false information posted by some accounts and recommended that this was performed in real time to reduce the danger of this information. Also, they discussed the lack of available datasets which help in the development of rumour classification systems.

Researchers have been doing studies on building and analysing COVID-19 Twitter datasets since the disease appeared in December 2019. So far, there are two different datasets which have been published recently related to Arabic and COVID-19 (Alqurashi et al., 2020) and (Haouari et al., 2020). The former collected 3,934,610 million tweets until April 15 2020, and the latter included around 748k tweets until March 31, 2020. These papers contain an initial analysis and statistical results for the collected tweets and some suggestions for future work, which include pandemic response, behavior analysis, emergency management, misinformation detection, and social analytics.

On the other hand, there are some datasets in English such as (Chen et al., 2020) and (Lopez et al., 2020). Also, there is a multilingual COVID-19 dataset containing location information (Qazi et al., 2020). This contains more than 524 million tweets, with 5.5 million Arabic tweets, posted over a period of three months since February 1, 2020. It focuses on determining the geolocation of a tweet which can help research with various different challenges, including identifying rumours.

Although the above studies have produced datasets related to COVID-19, they do not analyse them deeply using NLP methods. Previous studies representing earlier epidemics present good techniques and results, however none of them are related to Arabic tweets. Therefore, to assist PHOs in Arabic speaking countries there is an urgent need to analyse tweets related to COVID-19 using multiple Arabic NLP techniques.

## 3 Update Arabic Infectious Disease Ontology

With the recent appearance of COVID-19 as a new disease, there is need to update our Arabic Infectious Disease Ontology (Alsudias and Rayson,

2020), which integrates the scientific and medical vocabularies of infectious diseases with their informal equivalents used in general discourse. We collated COVID-19 information from the World Health Organization[6] and Ministry of health in Saudi Arabia. This included symptom, cause, prevention, infection, organ, treatment, diagnosis, place of the disease spread, and slang terms for COVID-19 and extended our ontology[7]. These terms were then used in our collection process.

## 4 Data Collection

We began collecting Arabic tweets about a number of infectious diseases from September 2019. Here in this paper, we analysed only the tweets related to COVID-19 from December 2019 to April 2020 (there are a few tweets between September and November, these are related to Middle East respiratory syndrome coronavirus, MERS-CoV[8]). We have collected approximately six million tweets in Arabic during this period. We obtained the tweets depending on three keywords (كورونَا, كرونَا, كوفِيد ١٩) which mean Coronavirus, a misspelling of the name of Coronavirus, and COVID-19 respectively in English. We collected the tweets weekly using Twitter API.

Next, we pre-processed the tweets through a pipeline of different steps:

- Manually remove retweets, advertisements, and spam.

- Filter out URLs, mentions, hashtags, numbers, emojis, repeating characters, and non-Arabic words using Python scripts[9].

- Normalize and tokenize tweets.

- Remove Arabic stopwords (Alrefaie, 2017).

After pre-processing, the resulting dataset was 1,048,575 unique tweets from the original 6,578,982 collected. Figure 1 shows the number of Arabic tweets about Coronavirus each week with specific dates highlighted to show government decisions on protecting the population from COVID-19 and other key dates for context.

---

[6] http://www.emro.who.int
[7] https://github.com/alsudias/ Arabic-Infectious-Disease-Ontology
[8] https://www.who.int/ news-room/fact-sheets/detail/

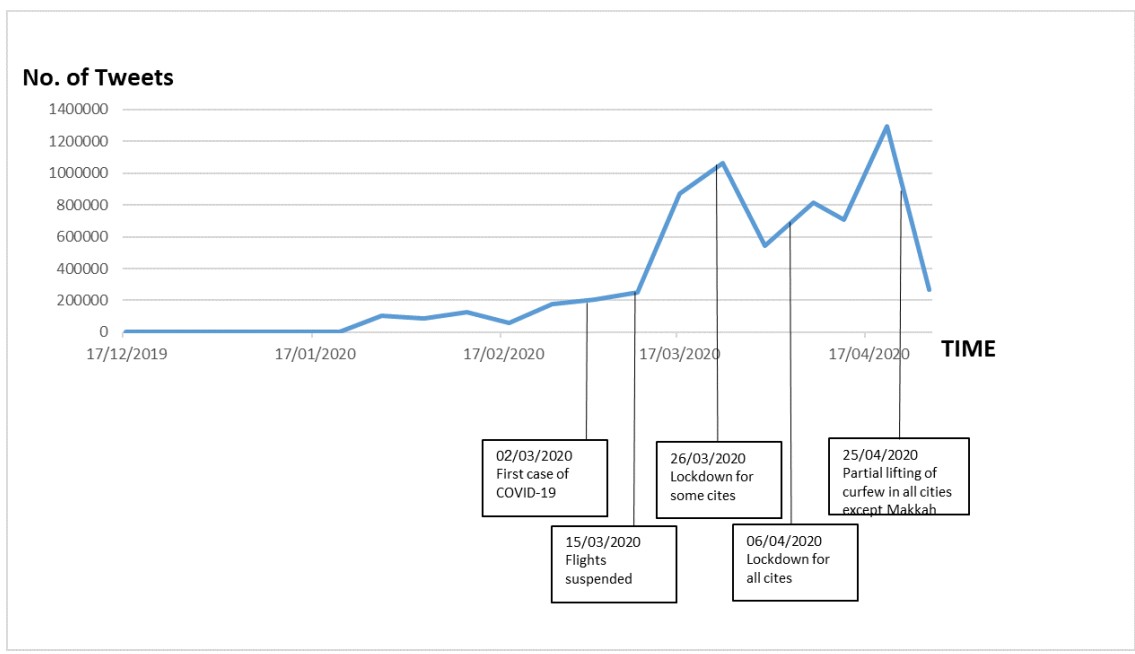

Figure 1: Number of Arabic tweets about Coronavirus

## 5 Methods

We performed three different types of analysis on the collected data. Firstly, in order to better understand the topics discussed in the corpus, we carried out a cluster analysis. Secondly, taking a sample of the corpus, we performed rumour detection. Finally, we extended our previous work to classify the source of tweets into five types of Twitter users which aims at helping to determine their veracity.

### 5.1 Cluster Analysis

To explore the topics discussed on Twitter during the COVID-19 epidemic in Saudi Arabia and other countries in the Arab World, we subjected the text of the tweets to cluster analysis. After pre-processing the tweets as described above, we used the N-gram forms (unigram, bigram, and trigram) of twitter corpus and clustered them using the K-means algorithm with the Python Scikit-learn 0.20.2 (Pedregosa et al., 2011) software and set the value of k, the number of clusters, to be five.

### 5.2 Rumour Detection

Following previous work (Zubiaga et al., 2018b), we applied a top-down strategy, which is where the set of rumours is identified in advance then the data is sampled to extract the posts associated with the previously identified rumours. In our dataset, out

of the one million tweets, we sampled 2,000 tweets to classify them for rumour detection. We manually labelled the tweets to create a gold standard dataset and then applied different machine learning algorithms in this part of our study.

#### 5.2.1 Labelling Guidelines

We manually labelled the tweets with 1, -1 , and 0 to represent false information, correct information, and unrelated content, respectively. Our reference point for deciding whether the content contained true or false information was based on the list issued by the Ministry of Health in Saudi Arabia [10] and is regularly updated (the last update applied for this study dates from 14 April 2020). Table 1 presents the list in both Arabic and English and Table 2 shows some example tweets for each label.

#### 5.2.2 Machine Learning Models

We applied three different machine learning algorithms: Logistic Regression (LR), Support Vector Classification (SVC), and Naïve Bayes (NB). To help the classifier distinguish between the classes more accurately, we extracted further linguistic features. The selected features fall into two groups: word frequency, count vector and TF-IDF, and word embedding based (Word2Vec and FastText). We used 10-fold cross validation to determine accuracy of the classifiers for this dataset, splitting the

middle-east-respiratory-syndrome-coronavirus-(mers-cov)
[9]https://github.com/alsudias
[10]https://www.moh.gov.sa

| Rumour in Arabic | Rumour in English |
|---|---|
| الحيوَانَات الَاليفة تنقل فيروس كورونَا. | Pets are transporters of Coronavirus. |
| البعوض نَاقل لّكورونَا. | Mosquitoes are transporters of Coronavirus. |
| الَاطفَال قد لَا يصَابون بفيروس كورونَا. | Children are not infected by Coronavirus. |
| كبَار السن فقط قد يتعرضون لمَخَاطر سيّئة من فيروس كورونَا. | Only old people may have a high risk of Coronavirus. |
| حرَارة الطقس أو برودته تقضي علَى الفيروس. | Hot or cold weather can kill the virus. |
| الغرغرة بَالمَاء و الملح تقضي علَى الفيروس. | Gargling with water and salt eliminates the virus. |
| يوجد خلطَات و أعشَاب تقي من الكورونَا. | There are some herbs that protect against from Coronavirus. |
| الفيروس لَا يبقَى علَى الآسطح. | The virus does not survive on surfaces. |

Table 1: List of rumours that appear during COVID-19 (source: Saudi Arabia Ministry of Health)

| Tweet in Arabic | Tweet in English | Label |
|---|---|---|
| سيكون هنَاك انحصَار لَانتشَار فيروس كورونَا مع بدَاية فصل الصيف خصوصَا في العَالم العربي نظرَا لَارتفَاع درجة الحرَارة. | There will be a decrease in the spread of the Corona virus at the beginning of the summer, especially in the Arab world, due to the high temperatures. | 1 (false) |
| الصحة: يعيش الفيروس و يرتكز بَالَاسَاس في الجهَاز التنفسي لذلك غير وَارد انتقَاله عن طريق الحشرَات أو من خلَال لدغة البعوض. | The Ministry of Health: A virus lives and is mainly concentrated in the respiratory system, so it is not likely to be transmitted by insects or by mosquito bites. | -1 (true) |
| اللّهم في هذي السَاعة المبَاركة نسآلك ان ترحمنَا و تبعد عنَا كل دَاء و بلَاء و قنَا شر الَامرَاض و الَاسقَام. وَاحفظ بلَادنَا و كَافة بلَاد المسلمين. | Oh God, in this blessed hour, We ask you to have mercy on us and keep away from us all disease and calamity, and protect us from the evil of diseases and sicknesses. Preserve our country and other Muslim countries. | 0 (unrelated) |

Table 2: Example tweets and our labelling system

entire sample into 90% training and 10% testing for each fold.

### 5.3 Source Type Prediction

We replicated a Logic Regression model from our previous study, which was useful for classifying tweets into five categories: academic, media, government, health professional, and public (Alsudias and Rayson, 2019). We used this LR model because it previously achieved the best accuracy (77%), and employed it here to predict the source of the COVID-19 tweets that we had already labelled in Section 5.2.

## 6 Results and Discussion

### 6.1 Cluster Analysis

Our cluster analysis of the five main public topics discussed in tweets content is as follows: (1)

disease statistics: the number of infected, died, and recovered people; (2) prayers: prayer asking God to stop virus; (3) disease locations: spread and location (i.e., name of locations, information about spread); (4) Advise for prevention education: health information (i.e., prevention methods, signs, symptoms); and (5) advertising: adverts for any product either related or not related to the virus. Figure 2 illustrates the five topics of COVID-19 tweets with examples.

For each cluster, the top terms by frequency are as follows: (1) disease statistics: case (حَالة), new (جديدة), and infection (اَصَابة); (2) prayers: Allah (اللّه), Oh God (اللّهم), and Muslims (المسلمين); (3) disease locations: Dammam (الدمَام), Riyadh (الرِياض), and Makkah (مكة); (4) Advise for prevention education: crisis (ازمة), spread (انتشَار), and pandemic (جَائمحة); (5) advertising: discount (خصم), coupon (كوبون), and code (كود).

We found that four of our categories (disease statistics, prayers, disease locations, and advise for prevention education) are similar to those found by Odlum and Yoon (2015) which are risk factors, prevention education, disease trends, and compassion. The marketing category is one of the topics in (Ahmed et al., 2017a) which discussed the topics in Twitter during the Ebola epidemic in the United States. Jokes and/or sarcasm is one of the categories that did not appear in our study but can be found in (Ahmed et al., 2017a) and (Ahmed et al., 2019), a thematic analysis study of Twitter data during H1N1 pandemic. This may be a result of more concern and panic from COVID-19 than other diseases during this period of time.

## 6.2 Rumour Detection

The result of our manual labelling process is 316 tweets label with 1 (false), 895 tweets label with 1 (true), and 789 tweets label with 0 (unrelated). Therefore, the false information represents about 15.8% (from 2,000 tweets) and around 26% (from 1,211 tweets, after removing the unrelated ones). In the study by Ortiz-Martínez and Jiménez-Arcia (2017), 61.3% (from 377 tweets) of data was classified as misinformation about Yellow Fever. It represented 32% (from 26,728 tweets) considered as rumours related to Zika Fever in Ghenai and Mejova (2017).

Figures 3, 4, and 5 show the accuracy, F1-score, recall, and precision on our corpus using LR, SVC, and NB algorithms with various feature selection approaches. The highest accuracy (84.03%) was achieved by the LR classifier with a count vector set of features and SVC with TF-IDF. Therefore, the count vector gives best result in LG and NB in all metrics results whereas with precision which achieves better results with TF-IDF 83.71% in LG and 81.28% in NB. while TF-IDF in SVC has the best results except recall which achieved 75.55% in count vector set features.

We also applied several word embedding based approaches but without obtaining good results. The accuracy ranges from 50% to 60% and the F1 score is around 40% on average. FastText models achieve better accuracy in SVC (54.89%) and NB (59.49%) than Word2Vec by approximately (5%). While Word2Vec shows the best result with LG 60.68% for accuracy, 49% for F1 and recall, and 65.97% in precision.

The word frequency based approaches have around a 20% better result than the word embedding ones. The reason for this is expected to be the dataset size and the specific domain of context (Ma, 2018). We assumed that the word embedding methods may achieve good results due to the importance of the relevant information around the word. For example, FastText can deal with the misspelling problem which is common in social media language style and improves word vectors with subword information (Bojanowski et al., 2017).

## 6.3 Source Type Prediction

The model predicts the source type for each of the tweets. Table 3 shows some examples of the tweets with predicted labels by the model. We focus on the result of the fake news content since they are of highest importance for the Public Health Organization. 30% (95 of 316) and 28% (91 of 316) of the rumour tweets are classified as written by a health professional and academic consequently. While only 12% (39 of 316) of them are predicted as written by the public. With this result, we find that the tweets containing false information quite often used the language style of academics and health professionals.

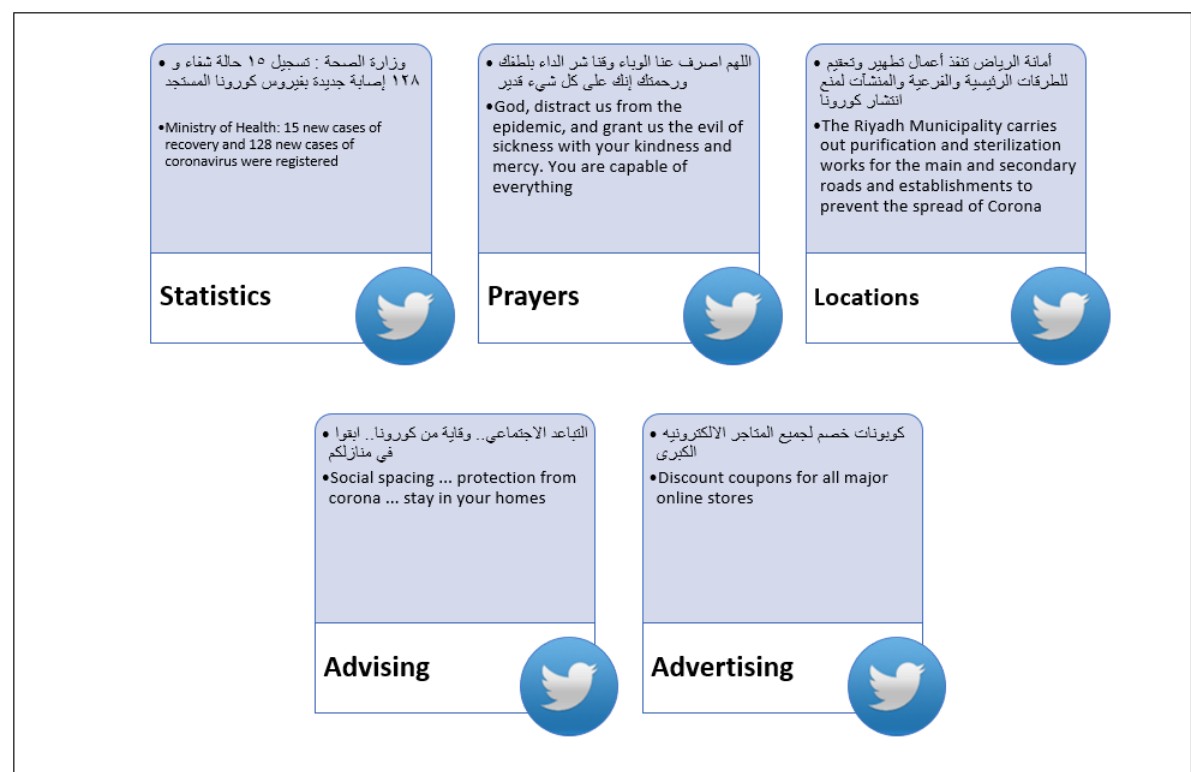

Figure 2: Examples of tweets in each cluster

| Tweet in Arabic | Tweet in English | Predicted Label |
|---|---|---|
| في قرَاءة علمية... يتوقع اندثَار الفيروس في ابريل بسبّب الحرَارة | In scientific reading ... the virus is expected to erode in April due to heat. | Academic |
| متحدث الصحة الحَالَات المُؤكدة حتَى الآن المصَابة بفيروس كورونا و معضمهَا لبَالغين | Health spokesman has confirmed cases so far infected with coronavirus, mostly for adults. | Media |
| يَا وزَارة رشوَا البعوض ، هو النَّاقل لفيروس كورونَا زَادت الأَصابَات مع انتشَار البعوض | Ministry of health please spray mosquitoes, as they are carriers of the Coronavirus, increased infections as mosquitoes spread. | Government |
| خبير صيني يُؤكد ان استنشَاق بخَار المَاء يقتل فيروس كورونَا | A Chinese expert confirms that inhaling water vapor kills coronavirus. | Health professional |
| علَاج الكرونَا بَالليّمون و الثوم "رابط يوتيوب" | Corona treatment with lemon and garlic "YouTube link". | Public |

Table 3: Some examples of false tweets from different source predicted labels

## 7 Conclusion and Future work

In this paper, we identified and analysed one million tweets related to the COVID-19 pandemic in the Arabic language. We performed three experiments which we expect can help to develop methods of analysis suitable for helping Arab World Governments and Public Health Organisations. Our analysis first identifies the topics discussed on social media during the epidemic, detects the tweets that contain false information,

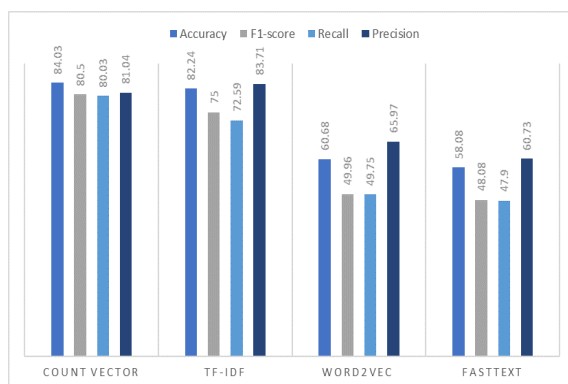

Figure 3: Results using Logistic Regression

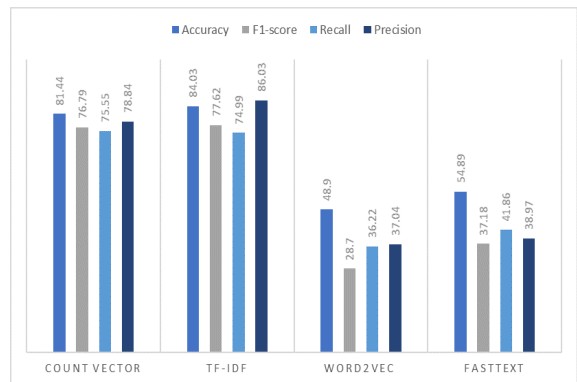

Figure 4: Results using Support Vector Classification

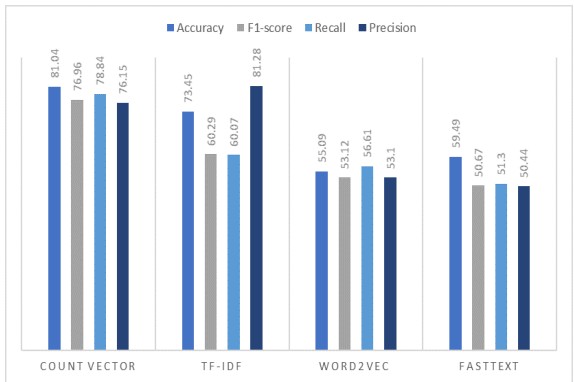

Figure 5: Results using Naïve Bayes

and predicts the source of the rumour related tweets based on our previous model for other diseases. The clustered topics are COVID-19 statistics, prayers for God, COVID-19 locations, advise for preventing education, and advertising.

Our second contribution is a labeled sample of tweets (2,000 out of 1 million) annotated for false information, correct information, and unrelated. To investigate the replicability and scalability of this annotation, we applied multiple Machine Learning Algorithms with different sets of features. The

highest accuracy result was 84% achieved by the LR classifier with count vector set of features and SVC with TF-IDF.

Finally, we also used our previous model to predict the source types of the sampled tweets. Around 60% of the rumour related tweets are classified as written by health professional and academics which shows the urgent need to respond to such fake news. The dataset, including tweet IDs, manually assigned labels for the sampled tweets, and other resources used in this paper are made freely available for academic research purposes [11].

There are clearly many potential future directions related to analysing social media data on the topics of pandemics. Since false information has the potential to play a dangerous role in topics related to health, there is a need to enhance and automate the automatic detection process supporting different languages beyond just English. Future potential directions include monitoring the spread of the disease by finding the infected individuals, defining the infected locations, or observing people that do not apply self isolation rules. Moreover, the analysis could proceed in an exploratory and thematic way such as discovering further topics discussed during the epidemic, as well as assisting governments and public health organisations in measuring people's concerns resulting from the disease.

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
