# OpenReview forum: "COVID-19 and Arabic Twitter: How can Arab World Governments and Public Health Organizations Learn from Social Media?"
_aclweb.org/ACL/2020/Workshop/NLP-COVID — NLP-COVID-2020_

### Official Review · AnonReviewer1 · 2020-06-22
**Worthy paper of limited scope that does what it sets out to do successfully and usefully.**

**Rating:** 8
**Confidence:** 4

**Review:**

The paper is well-written and organized. It analyzes a refined dataset of Arabic language tweets and successfully extracts helpful information from them. The information is potentially of great value to public health organizations who might not otherwise be collecting this data.

My only reservation about the paper is that it confines itself to Tweets in Arabic script, thus eliminating the many Tweets in the Arabic chat alphabet across North Africa in particular, in the various dialects found there. These would be of equal value for analysis, though they would require much more unpacking owing to differences in dialect and chat alphabet preferences. So perhaps that is a task for another team.

---

### Official Review · AnonReviewer3 · 2020-07-05
**Much-needed study and findings with potential for health-policy decision making before, during, and after infectious disease for the Arabic-Speaking countries**

**Rating:** 8
**Confidence:** 3

**Review:**

The paper is well-written with clear scope of research objective, methodology, hypotheses, findings, and implication.  Literature review is adequate and relevant.  The study was well conducted and clearly described.  It is also a much needed study for the Arabic-speaking countries to have machine-learning models to capture and analyze Arabic social media such as Twitter to discover important topics about COVID-19 and to classify the True information from False from reliable sources to influence the decision by the governments for public health.  The study, although small in terms of data sets, depth of classification, and different word-embedding models, sufficiently showed the value of and soundness of the approach to timely-information discovery from mining social media, especially for the Arabic language that presents different challenges from English social media.

---

### Official Review · AnonReviewer4 · 2020-07-05
**Methodologically sound contribution notable for its focus on Arabic social media**

**Rating:** 8
**Confidence:** 3

**Review:**

# [REVIEW] COVID-19 and Arabic Twitter:  How can Arab World Governments and Public Health Organizations Learn from Social Media?

ACL COVID Workshop 2020

5th July 2020

## SUMMARY
This paper describes work on processing 1 million Arabic language tweets from the Twitter API related to COVID-19 (i.e. containing the Arabic translations of the term “COVID”).  Analysis consisted of 3 phases.
First, k-means cluster analysis using ngrams was used to identify broad topics (e.g. disease statistics, prayers, disease location).  Second, automatically detecting rumours.  Second, a top-down strategy was used annotate 2,000 tweets according to pre-determined rumours (e.g. hot weather kills covid) achieving an accuracy of 84% (logistic regression).  Third, a logistic regression model developed in a prior study was used to classify tweets by source (i.e. academic, media, government, health professional, and public).  This classifier was applied to the labelled data generated in step 2, when it was discovered that tweets containing false information and rumours, often adopt the language style of academics/official discourse, creating further challenges for health communication.

This is a (generally) well-written, clear paper.  It is not methodologically innovative, but it is nicely executed, and the fact that it focuses specifically on Arabic language tweets makes it a valuable contribution.

## COMMENTS

* Why cluster analysis over, say, biterm topic modelling?
* How was the appropriate number of clusters determined?

---

### Decision · Program_Chairs · 2020-07-06

**Decision:**

Accept

**Comment:**

Thank you for your submission.

The reviewers have given positive feedback on this work, and we are pleased to include it for presentation in the Workshop on Thursday (5:30-9:30pm PDT).

Please plan on a 10-minute video presentation; probably pre-recorded is best but live will be an option as well.

We look forward to hearing more about this work.

Apologies for the late decision!